# A meta-epidemiological assessment of transparency indicators of infectious disease models

**Emmanuel A. Zavalis** [1,2☯], **John P. A. Ioannidis** [1,3☯] *

**1** Meta-Research Innovation Center at Stanford (METRICS), Stanford University, Stanford, California, United States of America, **2** Department of Learning, Informatics, Management and Ethics, Karolinska Institutet, Solna, Stockholm, Sweden, **3** Departments of Medicine, of Epidemiology and Population Health, of Biomedical Data Science, and of Statistics, Stanford University, Stanford, California, United States of America

☯ These authors contributed equally to this work.
* jioannid@stanford.edu

**Data Availability Statement:** The protocol for this study can be found at doi:10.17605/OSF.IO/JGWVK. The code and data is available at https://github.com/zavalis/transparency-indicators-inf-dis-models.

## Abstract

Mathematical models have become very influential, especially during the COVID-19 pandemic. Data and code sharing are indispensable for reproducing them, protocol registration may be useful sometimes, and declarations of conflicts of interest (COIs) and of funding are quintessential for transparency. Here, we evaluated these features in publications of infectious disease-related models and assessed whether there were differences before and during the COVID-19 pandemic and for COVID-19 models versus models for other diseases. We analysed all PubMed Central open access publications of infectious disease models published in 2019 and 2021 using previously validated text mining algorithms of transparency indicators. We evaluated 1338 articles: 216 from 2019 and 1122 from 2021 (of which 818 were on COVID-19); almost a six-fold increase in publications within the field. 511 (39.2%) were compartmental models, 337 (25.2%) were time series, 279 (20.9%) were spatiotemporal, 186 (13.9%) were agent-based and 25 (1.9%) contained multiple model types. 288 (21.5%) articles shared code, 332 (24.8%) shared data, 6 (0.4%) were registered, and 1197 (89.5%) and 1109 (82.9%) contained COI and funding statements, respectively. There was no major changes in transparency indicators between 2019 and 2021. COVID-19 articles were less likely to have funding statements and more likely to share code. Further validation was performed by manual assessment of 10% of the articles identified by text mining as fulfilling transparency indicators and of 10% of the articles lacking them. Correcting estimates for validation performance, 26.0% of papers shared code and 41.1% shared data. On manual assessment, 5/6 articles identified as registered had indeed been registered. Of articles containing COI and funding statements, 95.8% disclosed no conflict and 11.7% reported no funding. Transparency in infectious disease modelling is relatively low, especially for data and code sharing. This is concerning, considering the nature of this research and the heightened influence it has acquired.

**Funding:** EAZ visiting fellowship at Stanford was partly funded by the foundation 'Carl Erik Levins Stiftelse' and by the Meta-Research Innovation Center at Stanford (METRICS). The funders had no role in study design, data collection and analysis, decision to publish, or preparation of the manuscript.

**Competing interests:** I have read the journal's policy and the authors of this manuscript have no competing interests.

## Introduction

A large number of infectious disease-related models are published in the scientific literature and their production and influence has rapidly increased during the COVID-19 pandemic. Such models can inform and shape policy, and have also been the subject of much debate [1–4], surrounding a range of issues, including their questionable predictive accuracy and their transparency [5–7].

Sharing of data and of code is totally indispensable for these models to be properly evaluated, used, reused, updated, integrated, or compared with other efforts. Without being able to rerun a model, it resembles a black box where blind trust is requested on its function and credibility. Moreover, other features of transparency, such as declaration of funding and of potential conflicts of interest (COI) are also important to have since many of these models may be very influential on deciding policy with major repercussions. Another feature of transparency that may aid reproducibility and trust in these models sometimes is the registration of their protocols, ideally in advance of their conduct. Registration is concept that receives increasing attention in many scientific fields [8–10] as a safeguard of trust. Registration may not be easy or relevant to have for many mathematical models, especially those that are exploratory and iterative [5]. However, it may be feasible and desirable to register protocols about models in some circumstances [5].

There have previously been empirical evaluations of research practices, including documentation and transparency in subfields of mathematical modeling [11–13] that have shown that data and code/algorithm sharing has improved somewhat over time but that it still remains suboptimal. Yet, to our knowledge, in the field of infectious disease modelling there has been no comprehensive, large-scale analysis of such transparency and reproducibility indicators. It would be of interest to explore the state of transparency in this highly popular field, especially in the context of the rapid and massive adoption of mathematical models during the COVID-19 pandemic. Therefore, we decided to evaluate infectious disease modeling studies using large-scale algorithmic extraction of information on several transparency and reproducibility indicators (code sharing, data sharing, registration, funding, conflicts of interest). We compared these features in articles published before and during the pandemic (in 2019 and 2021, respectively) and in articles on COVID-19-related models and models related to other infectious diseases.

## Materials and methods

This study is a meta-epidemiological survey of transparency indicators present in four common types of infectious disease models (compartmental, spatiotemporal, agent-based/individual based and time-series) indexed in the PubMed Central Open Access (PMC OA) Subset of PubMed. The study is reported using the STROBE guidelines [14]. The code needed for the analysis of our data used R [15] and Python [16].

### Search and screening

We developed a search strategy to identify papers published from 2019 and 2021 in English in PMC OA subset that included models of infectious diseases: model*[tiab] OR forecast*[tiab] OR predict*[tiab]) AND (SIR-models[tiab] OR SIR[tiab] OR SIRS[tiab] OR SEIR[tiab] OR SEIR-model[tiab] OR SIRS-model[tiab] OR agent-based[tiab] OR spatiotemporal[tiab] OR nowcast[tiab] OR backprojection[tiab] OR "traveling waves"[tiab] OR (time series[tiab] OR time-series[tiab])) NOT (rat model*[ti] OR murine model*[ti] OR animal model*[ti] OR mouse model*[ti] OR primate model*[ti]) AND (infect* OR transmi* OR epidem*. The model

types that were included were compartmental models, spatiotemporal models, agent-based/ individual-based models and time series models. They were defined as follows:

(i) Compartmental models assign subsets of the population to different classes according to their infection status (e.g., susceptible exposed, recovered etc.) and models the population parameters of the disease according to assumed transmission rates between these subsets [17].

(ii) Spatiotemporal models explore and predict the temporal and geographical spread of infectious diseases (usually using geographic time series data).

(iii) Agent-based/ individual based models are computer simulations of the interaction of agents with unique attributes regarding spatial location, physiological traits and/or social behavior [18, 19]. Finally,

(iv) Time-series models other than spatiotemporal were also included that use trends in number of infected or deaths or any other parameter of interest to predict future trends and numbers of spread [20].

We excluded clinical predictive, prognostic, and diagnostic models and included only models of infectious agents that can infect humans (i.e. both zoonotic diseases as well as diseases exclusive to humans). All screening and analysis was conducted by EAZ in two eligibility assessment rounds. In the first round, eligibility was assessed based on the title and abstract; in the second where the model type and disease type was extracted, eligibility was also assessed by perusing the article in more depth. After this round, in unclear cases EAZ consulted JPAI and these cases were settled with discussion.

## Data extraction

For each eligible study, we extracted information on the model type and disease type manually. For model type, whenever cases came up that were not clear-cut EAZ and JPAI conferred as to what category was sensible. Some phylogenetic models were included and classified as spatiotemporal if they had spatiotemporal aspects. When there were multiple model types in a single paper it was classified as 'Multiple'. For disease, we used categories defined based on the infectious agent of interest that was studied. The "Unspecified" category included studies not mentioning a specific infectious agent but a clinical syndrome (i.e. urinary tract infection or pneumonia etc.), the "General (theoretical models)" category included studies that didn't model a specific disease (i.e. a theoretic pandemic). Finally, where multiple diseases were mentioned, the papers were categorised in a separate category as 'Multiple different agents' (i.e. HIV and tuberculosis). Where vectors of diseases such as mosquitos were modelled to predict spread of multiple diseases, we classified the disease as 'Vector'.

For each eligible article we used PubMed to extract information on metadata that included PMID, PMCID, publication year, journal name and the R package *rtransparent* [21] to extract the following transparency indicators: (i) code sharing (ii) data sharing (iii) (pre-)registration, (iv) COI and (v) funding statements.

*rtransparent* searches through the full text of the papers for specific words or phrases that strongly suggest that the aforementioned transparency indicators are present in that particular paper. The program uses regular expressions to adjust for variations in expressions. For example, to identify code sharing, rtransparent looks for "code" and "available" as well as the repository "GitHub" and its variations, and in a paper selected [22] from our dataset it finds the following: "the model and code for reproducing all figures in this manuscript from model output are publicly available online (https://github.com/bdi-pathogens/openabm-covid19-model-paper)"

The approach has been previously validated and tested in Serghiou et al. [21] across the entire biomedical literature and has a positive predictive value (PPV) of 88.2% (81.7%-93.8%) and negative predictive value (NPV) of 98.6% (96.2–99.9%) for code sharing; 93.0% (88.0%-97.0%) and 94.4% (89.1%-97.0%) for data sharing, 92.1% (88.3–98.6%) and 99.8% (99.7%-99.9%) for registration, 99.9% (99.7%-100.0%) and 96.8% (94.4%-99.1%) for COI disclosure and 99.7% (99.3%-99.9%) and 98.1% (96.2%-99.5%) for funding disclosures.

To further validate the performance of the algorithms in detecting code sharing and data sharing reliably, a random sample of 10% of publications that the algorithm identified as sharing code and 10% of those that the algorithm identified as sharing data were manually assessed looking into whether the statements indeed represented true sharing. All papers that were identified by the algorithm to have registration were assessed manually to verify whether registration had been performed. After a suggestion by a reviewer, we also examined manually random samples of 10% of the publications that were found by the algorithm to have not satisfied each indicator. The corrected proportion C(i) of publications satisfying an indicator i was obtained by U(i) × TP + (1 − U(i)) × FN, where U(i) is the uncorrected proportion detected by the automated algorithm, TP is the proportion of true positives (proportion of those manually verified to satisfy the indicator among those identified by the algorithm as satisfying the indicator, and FN is the proportion of false negatives (proportion of those manually found to satisfy the indicator among those categorized by the algorithm not to satisfy the indicator). Moreover, random sample of 10% of papers that were found to contain a COI statement and 10% of those found to include a funding statement were assessed manually to see not only whether such statements were indeed present, but also to assess how many of them contain actual disclosures of specific conflicts or funding sources, respectively, and not just a statement that there are no COIs/funding, e.g. 'There is no conflict of interest', No funding was received' or 'Funding disclosure is not applicable'. Finally, a random sample of 10% of the negatives for COI and funding were also manually assessed.

## Statistical analysis

The primary outcome studied was the percentage of papers that include each of the transparency indicators. We considered three primary comparisons that were conducted using Fisher's exact tests.

- All publications in 2019 to all in 2021 (to assess if there is improvement over time)

- All non-COVID-19 publications in 2019 to the non-COVID-19 publications in 2021 (to assess if there is improvement over time for non-COVID-19 publications)

- 2021 COVID-19 publications to 2021 non-COVID-19 ones (to assess if COVID-19 papers differ in transparency indicators versus non-COVID-19 papers).

Subsequently we also explored whether other factors may have correlated with the transparency indicators using Fisher's exact tests to see whether there was any statistically significant association (significance level set at 0.005 [23]) when comparing model types, year, disease modelled, as well as journal separately. We had pre-specified that whenever any statistically significant results were found, we would conduct multivariable logistic regressions as well using the transparency indicators as the dependent variable. This was to see if any of the covariates, which for the regression to be able to converge had to be larger groups, were alone associated with our outcome variables. The covariates used were therefore year and disease combined (2019 (baseline), 2021 non-COVID-19, 2021 COVID-19), Journal (PLoS One, Scientific Reports, International Journal of Environmental Research and Public Health, Other

(baseline)) and the type of model (with the compartmental models used as a baseline). Statistical significance was claimed for p<0.005 and p-values between 0.005 and 0.05 are considered suggestive, as previously proposed [23].

## Deviations from the protocol

We deviated from the protocol in that we didn't perform chi-square tests due to too low counts in some variables rendering it unreliable, therefore we decided to conduct these analyses using Fisher's exact tests instead of chi-square tests. The 10% manual assessment of a random sample of articles with COI and funding statements was added post hoc, when we realized that many articles could have such statements, but they might simply state that there was no COI and/or no funding.

## Results

### Study sample

We screened 2903 records in their titles and abstracts according to the eligibility criteria. 1340 papers were excluded due to ineligibility in the primary survey leaving 1563 records for further scrutiny. 58 were excluded during the second round of screening, i.e., during retrieval of information on model type and disease and 167 were excluded for not being part of the PMC OA subset (Fig 1).

**Characteristics of eligible papers.** Of the 1338 eligible papers (Table 1), 216 had been published in 2019 and 1122 in 2021. 818 (61.1%) were COVID-19 papers and the second largest group contained 130 (9.7%) publications and was the group of General (theoretical models). More than 70 different diseases had altogether been modelled in the eligible publications. The model types were more evenly distributed with the most common model type being compartmental models (N = 511, 39.2%) and time series models (N = 337, 25.2%).

### Transparency indicators

Table 2 shows the transparency indicators overall and in the three main categories based on year and COVID-19 focus. We found that based on the text mining algorithms 288 (21.5%)

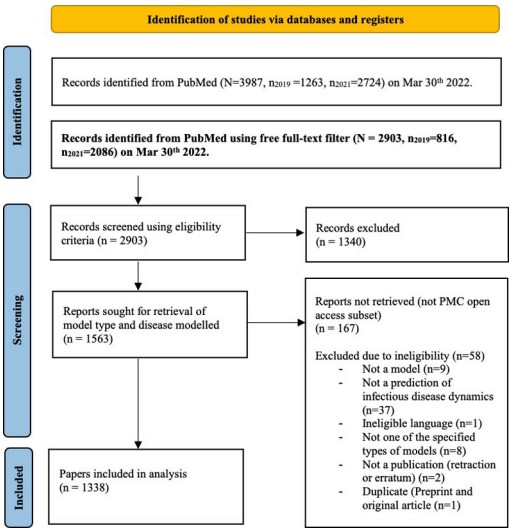

**Fig 1. Flow chart for study selection.**

**Table 1. Characteristics of eligible studies.**

| | 2019 | 2021 non-COVID-19 | 2021 COVID-19 | All publications |
|---|---|---|---|---|
| | N (%) | N (%) | N (%) | N (%) |
| | 216 articles | 304 articles | 818 articles | 1338 articles |
| **Type of model** | | | | |
| Compartmental | 26 (12.0) | 91 (29.9) | 394 (48.0) | 511 (39.2) |
| Time series | 80 (37.0) | 82 (27.0) | 175 (21.4) | 337 (25.2) |
| Spatiotemporal | 78 (36.1) | 90 (29.6) | 111 (13.6) | 279 (20.9) |
| Agent-based | 31 (14.4) | 37 (12.2) | 118 (14.4) | 186 (13.9) |
| Multiple | 1 (0.5) | 4 (1.3) | 20 (2.4) | 25 (1.9) |
| **Type of disease** | | | | |
| COVID-19 | 0 (0) | 0 (0) | 818 (100) | 818 (61.1) |
| General | 33 (15.3) | 97 (31.9) | 0 (0) | 130 (9.7) |
| Influenza illnesses | 20 (9.3) | 20 (6.6) | 0 (0) | 40 (3.0) |
| Malaria | 15 (6.9) | 22 (7.2) | 0 (0) | 37 (2.8) |
| Dengue | 15 (6.9) | 20 (6.6) | 0 (0) | 35 (2.6) |
| Others | 133 (61.6) | 145 (48) | 0 (0) | 278 (20.8) |
| **Journal** | | | | |
| PLoS One | 26 (12.0) | 27 (8.9) | 62 (7.6) | 115 (8.6) |
| Sci Rep | 20 (9.3) | 19 (6.3) | 52 (6.4) | 91 (6.8) |
| Int J Environ Res Public Health | 15 (6.9) | 21 (6.9) | 27 (3.3) | 63 (4.7) |
| BMC Infect Dis | 16 (7.4) | 12 (3.9) | 10 (1.2) | 38 (2.8) |
| PLoS Negl Trop Dis | 11 (5.1) | 22 (7.2) | 0 (0) | 33 (2.5) |
| PLoS Comput Biol | 10 (4.6) | 10 (3.3) | 9 (1.1) | 29 (2.2) |
| BMC Public Health | 6 (2.8) | 9 (3.0) | 13 (1.6) | 28 (2.1) |
| Chaos Solitons Fractals | 0 (0) | 5 (1.6) | 20 (2.4) | 25 (1.9) |
| Others | 112 (52.0) | 179 (58.9) | 625 (76.4) | 916 (68.5) |

articles shared code, 332 (24.8%) shared data, 6 (0.4%) used registration, and 1197 (89.5%) and 1109 (82.9%) contained a COI and funding statement, respectively. 919 (68.7%) of publications shared neither data nor code, while 199 (14.9%) of all papers shared both data and code.

We found no differences between years and between COVID-19 and non-COVID-19 papers in terms of probability of sharing data, registration, or mentioning of COIs.

**Table 2. Key transparency indicators overall and per year/COVID-19 focus.**

| N = 1338 | Code sharing | Data sharing | Registration | COI | Funding |
|---|---|---|---|---|---|
| | N (%) | N (%) | N (%) | N (%) | N (%) |
| **Overall** | 288 (21.5) | 332 (24.8) | 6 (0.4) | 1197 (89.5) | 1109 (82.9) |
| 2019 | 38 (17.6) | 59 (27.3) | 3 (1.4) | 197 (91.2) | 202 (93.5) |
| 2021 | 250 (22.3) | 273 (24.3) | 3 (0.3) | 1000 (89.2) | 907 (80.8) |
| COVID-19 | 207 (25.3) | 199 (24.3) | 0 | 730 (89.2) | 635 (77.6) |
| non-COVID-19 | 43 (14.1) | 74 (24.3) | 3 (1) | 270 (88.8) | 272 (89.5) |
| **Fisher's exact test (p-values)** | | | | | |
| *2019 vs 2021* | 0.15 | 0.35 | 0.06 | 0.45 | $1.0 \times 10^{-6}$ |
| *2019 vs 2021 non-COVID-19* | 0.33 | 0.48 | 0.70 | 0.46 | 0.12 |
| *2021 non-COVID-19 vs. COVID-19* | $5.1 \times 10^{-5}$ | 1 | 0.02 | 0.83 | $3.5 \times 10^{-5}$ |

COI: conflicts of interest

COVID-19 papers were more likely to share their code openly than the non-COVID-19 publications from the same year (14.1% v. 25.3%, p = $5.1 \times 10^{-5}$), and they were less likely to report on funding compared with non-COVID-19 papers in the same year (p = $3.5 \times 10^{-5}$). This led to an overall lower percentage of papers reporting on funding in 2021 compared with 2019 (p = $1.0 \times 10^{-6}$).

## Other correlates of transparency indicators

As shown in Table 3, data sharing varied significantly across journals, e.g. it was 54.8% in PLoS One, but 12.7% in International Journal of Environmental Research and Public Health. Code sharing varied significantly across diseases, e.g. it was most common for dengue and least common for malaria (34.3% v 5.4%); and it varied significantly among types of models, (highest in agent-based models with 33.9% of publications sharing code). Registration was uncommon in all subgroups. COI disclosures were most common in dengue and least common in general models and they also varied by type of model (least common in compartmental models). Funding information was most commonly disclosed in dengue models and least commonly disclosed in general models; it also varied by type of model (being lowest in compartmental models); and by journal.

Multivariable regressions (not shown) showed similar results. Code sharing was more common in COVID-19 models (OR 1.69 (1.13, 2.55) compared to the 2019 baseline) and in agent-based models (OR 2.15 (1.47, 3.14) using compartmental models as the baseline). Data sharing was more common in spatiotemporal (OR 1.90 (1.33, 2.73) and agent-based models (OR 1.80

**Table 3. Key transparency indicators per disease type, model type, and journal.**

| | Code sharing | Data sharing | Registration | COI | Funding |
|---|---|---|---|---|---|
| | N (%) | N (%) | N (%) | N (%) | N (%) |
| **Disease modelled** | | | | | |
| *p (Fisher's exact test)* | $7.4 \times 10^{-6}$ | 0.47 | 0.001 | 0.01 | $2.8 \times 10^{-10}$ |
| COVID-19 | 207 (25.3) | 199 (24.3) | 0 (0) | 730 (89.2) | 635 (77.6) |
| General (theoretical model) | 31 (23.8) | 34 (26.2) | 0 (0) | 94 (72.3) | 108 (83.1) |
| Influenza illnesses | 6 (15) | 10 (25) | 0 (0) | 38 (95) | 39 (97.5) |
| Malaria | 2 (5.4) | 7 (18.9) | 2 (5.4) | 37 (100) | 35 (94.6) |
| Dengue | 12 (34.3) | 13 (37.1) | 0 (0) | 35 (100) | 35 (100) |
| Other diseases | 30 (10.8) | 69 (24.8) | 4 (1.4) | 263 (94.6) | 257 (92.4) |
| **Type of model** | | | | | |
| *p (Fisher's exact test)* | 0.001 | 0.006 | 0.15 | $<1 \times 10^{-7}$ | 0.008 |
| Compartmental | 104 (20.4) | 103 (20.2) | 0 (0) | 419 (82) | 405 (79.3) |
| Time Series | 65 (19.3) | 81 (24) | 2 (0.6) | 319 (94.7) | 276 (81.9) |
| Spatiotemporal | 52 (18.6) | 84 (30.1) | 3 (1.1) | 263 (94.3) | 247 (88.5) |
| Agent-based | 63 (33.9) | 58 (31.2) | 1 (0.5) | 173 (93) | 161 (86.6) |
| Multiple | 4 (16) | 6 (24) | 0 (0) | 23 (92) | 20 (80) |
| **Journal** | | | | | |
| *p (Fisher exact)* | 0.15 | $1.7 \times 10^{-12}$ | 0.11 | $2.5 \times 10^{-12}$ | $3.4 \times 10^{-14}$ |
| PLoS One | 30 (26.1) | 63 (54.8) | 1 (0.9) | 115 (100) | 115 (100) |
| Sci Rep | 23 (25.3) | 21 (23.1) | 1 (1.1) | 91 (100) | 70 (76.9) |
| Int J Environ Res Public Health | 8 (12.7) | 7 (11.1) | 1 (1.6) | 63 (100) | 63 (100) |
| Other journals | 227 (21.2) | 241 (22.5) | 3 (0.3) | 928 (86.8) | 861 (80.5) |

COI: conflicts of interest

(1.21, 2.66)) compared to the baseline and also depended substantially on the journal (with PLoS One having OR 4.22 (2.84, 6.32) compared to the baseline of all journals but the top 3). We did not perform multivariable regressions for the presence of COI and funding statements, since these depended almost entirely on the journal (several journals had 100% frequency of having a placeholder for such statements). Registration was too uncommon to subject to multivariable analysis.

## Manual validation

We also checked a random sample of 29 (10%) of papers that were found to be sharing code, 33 (10%) of those sharing data, and all 6 that were registered. Of these, 24/29 (82.8%) actually shared code, 29/33 (87.9%) actually shared data and 5/6 (83.3%) were indeed registered. The papers that used registration were two malaria models [24, 25], one vector model [26] (which focused on malaria vectors) one polio (Sabin 2 virus [27]) model and one rotavirus model [28]. The majority were from 2021 [24, 26, 27] and were also malaria models (two malaria and one vector that was essentially malaria [24–26]). A majority was also classified as spatiotemporal [24–26]. We also checked a random sample of 10% of the negatives i.e. the ones that were classified as non-transparent and found that 133/133 (100%) weren't registered, 95/106 (89.6%) didn't share code and 75/101 (74.3%) didn't share data. Therefore, the corrected estimates of the proportions of publications sharing code and sharing data were (0.215 × 0.828) + (0.785 × 0.104) = 26.0% and (0.248 × 0.879) + (0.752 × 0.257) = 41.1%, respectively. The modest number of false-negatives for detecting data sharing through the text mining algorithms reflected mostly situations where it was mentioned that the data can be downloaded through a link, or the reference was in a figure, or the phrasing was interwined and difficult to separate effectively by the text mining algorithm.

Finally, of the 120 articles (10%) that text mining found that they contained a COI statement, there was indeed a placeholder for this statement in all articles, but the vast majority of the statements (115 (95.8%)) disclosed no conflict at all. Of the 111 (10%) articles where text mining found that they contained a funding statement, all of them had indeed such a statement, but 13 (11.7%) stated that they had no funding. Examining a random sample of 10% of the negatives regarding COI and funding disclosures we found that 19/23 (82.6%) of funding disclosures and 14/14 (100%) of COI disclosures were true negatives.

## Discussion

Analysing 1338 recent articles from the field of infectious disease modelling we found that based on previously validated text mining algorithm less than a quarter of these publications shared code or data, and only 14% shared both. Adding further validation through manual evaluation suggested that data sharing may be modestly more common, but still the majority of these publications did not share their data. This is concerning since it does not allow other scientists to check the models in any depth and it also limits their further uses. Moreover, registration was almost nonexistent. On a positive note, the large majority of models did provide some information on funding and COIs. However, the vast majority of COI statements simply said that there was no conflict. Furthermore, we saw no major differences between 2019 and 2021. COVID-19 and non-COVID-19 papers showed largely similar patterns for these transparency indicators, although the former were modestly more likely to share code and modestly less likely to report on funding. There were some differences for some of the transparency indicators across journals, model types and diseases.

Jalali et al. [11] analysed 29 articles on COVID-19 models in 2020 and found that 48% shared code, in 60% data was shared, whilst 80% contained a funding and COI disclosure

respectively. Our findings show much lower rates of code sharing and data sharing. The Jalali et al. sample was apparently highly selective as it focused on the most referenced models among a compilation of models by the US Centers for Disease Control [29]. In another empirical assessment of the reproducibility of 100 papers in simulation modelling in public health and health policy published over half a century (until 2016) and covering all applications (not just infectious diseases), code was available for only 2% of publications [30]. Finally, in an empirical evaluation in decision modelling by Emerson et al. [13], when the team tried to get authors of papers to share their code 7.3% of simulation modelling researchers responded and in the end only 1.6% agreed to share their code. This suggests that infectious disease models are not doing worse than other mathematical models, and may be doing even substantially better, but there is still plenty of room for improvement in sharing practices.

There have been many initiatives for improving sharing code and better documentation in the modelling community [31–34] as well, repositories for COVID-19 models [35, 36]. The modelling community including COVID-19 [37] modelling has had multiple calls for transparency and the debate of reproducibility has been ongoing for decades [38–40]. Several journals have tried to take steps in enhancing reproducibility. For example, Science changed their policy for code and data sharing to make both essentially mandatory [41]. However, Stodden et al. [42] found no clear improvement after such interventions. Models are published in a vast array of journals and sharing rate as well as reporting and documentation requirements tend to be highly journal specific.

The frequency of code and data sharing in our sample was higher than what was documented for the general biomedical literature that was assessed in Serghiou et al. [21] using the same algorithm. COI and funding disclosures were almost equally common. On the other hand, we observed a ten-fold lower registration rate in our sample compared with the overall biomedical literature, which may reflect the difficulty of registering models and the lack of sufficient sensitization of the field to this possibility [5]. We found that essentially 5 of our studies were registered (after validating the initial 6 that we found). Realising that registration may be difficult and even impossible for a large portion of models (exploratory models for instance) [5], it would still be advisable to register confirmatory studies of models that are destined to be used for policy to reduce the "vibration of effects" (the range of possible results obtained with multiple analytical choices) [43, 44]. Otherwise, promising output or excellent fit may in reality be due to bias alone. When the stakes are high and wrong decisions may have grave implications, more rigor is needed.

The rates of COI and funding disclosures are satisfactory on face value, considering they both are above 80% both in our sample and across other empirical assessments [11, 21, 45]. This may also be due to the fact that both these types of disclosures have been introduced into many journal's routinely published items and there is a standard placeholder for them. Typically journals mandate a COI and funding statement. However, the fidelity and completeness of these statements is difficult to probe. We cannot exclude that undisclosed COIs may exist. Our random sample validation found that the COI disclosures almost never mentioned any conflict. Given the policy implications of many models, especially in the COVID-19 era, this pattern may represent under-reporting of conflicts. Funding disclosures were more informative with only 12% stating no funding, but even then unstated sources of funding cannot be excluded.

## Limitations

There are limitations in our evaluation. Our sample focused on the PubMed Central Open Access subset and not all PubMed-indexed papers. It is unclear if non-open access papers may

be less likely to adopt sharing practices. If so, the proportion of sharing in the total infectious disease modeling literature may be over-estimated. Moreover, much of the COVID-19 literature was not published in the indexed peer-reviewed or indexed literature and therefore may have evaded our evaluation (even though some preprints are indexed in PubMed). If anything, this evading literature may have even less transparency.

Second, we used a text-mining approach which has been extensively validated across the entire biomedical literature, but the algorithms may have different performance specifically in the infectious disease modeling field. Nevertheless, in-depth evaluation of random samples of papers suggests that identification of these indicators is quite accurate and false positives are uncommon and well balanced by an almost equal number of false negatives for code sharing. For data sharing, the manual validation found a modest number of publications that had shared data but were not picked as sharing by the algorithm. Therefore, data sharing may occur modestly more frequently than suggested by the automated algorithm, but even then the majority of the publications in this field do not share their data.

Third, the presence of a data sharing or code sharing statement doesn't promise full functionality and the ability to fully reuse the data and code. This can only be decided after spending substantial effort to bring a paper to life based on the shared information. For COI and funding statements, we also only established their existence, but did not appraise in depth the content of these statements, let alone their veracity. Evaluations in other fields suggests that many COIs are undisclosed and funding information is often incomplete [46–48].

Fourth, using only one main reviewer for screening for eligibility may have introduced some errors in the selection of specific studies that were included or not in our analysis. However, identification of eligible studies is quite straightforward given our eligibility criteria and any ambiguous cases were also discussed with the second author. There were a few studies that did not fit squarely in our pre-determined categories, but their number is too small to affect the overall results.

Finally, we only assessed a sample that is drawn from two calendar years that are not very far apart, thus major changes might not have been anticipated at least for non-COVID-19 models. Nevertheless, 2021 was a unique year with a pandemic which of course affected the field not merely through inflation of publications [49] but also through specific funder and governmental initiatives and incentives. Therefore, only time will tell if any of the COVID-19 impact on the scientific literature will be long-lasting and if it may also affect the landscape of mathematical modeling in general after the pandemic phases out.

## Conclusions

We found that in the highly influential field of infectious disease modeling that relies as much on its assumptions and underlying code and data, transparency and reproducibility have large potential for improvement. Yet, there is a growing literature of recommendations and tutorials for researchers and other stakeholders [50–53], plus the EPIFORGE guidelines [54] which are guidelines for the reporting of epidemic forecasting and prediction research. They all explicitly urge for code sharing, and data sharing and transparency in general. The current lack of transparency may cause problems in the use, reuse, interpretation, and adoption of these models for scientific or policy activities. It also hinders evidence synthesis and attempts to build on previous research to facilitate progress within the field. Improved transparency and reproducibility may help reinforce the legacy of this important field. It can be argued that a mathematical model should not be taken seriously, especially for influential inferences and decisions, without the underlying code and data sources made public. This includes models published by academic journals or unpublished ones that are being used nevertheless to guide health

policies or other decisions. One might even suggest banning the publication of models that do not share their data and code. Pre-registration also is highly desirable, when pertinent, and for some targeted uses of models, e.g. making claims for future predictions, it should become a normal expectation.

## Acknowledgments

We thank Professor Carl Johan Sundberg for his thoughtful feedback to the paper.

## Author Contributions

**Conceptualization:** Emmanuel A. Zavalis, John P. A. Ioannidis.

**Data curation:** Emmanuel A. Zavalis.

**Formal analysis:** Emmanuel A. Zavalis, John P. A. Ioannidis.

**Investigation:** Emmanuel A. Zavalis, John P. A. Ioannidis.

**Methodology:** John P. A. Ioannidis.

**Supervision:** John P. A. Ioannidis.

**Writing – original draft:** Emmanuel A. Zavalis, John P. A. Ioannidis.

**Writing – review & editing:** Emmanuel A. Zavalis, John P. A. Ioannidis.

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
