## [Decision Letter · Decision Letter 0]

7 Jul 2022

PONE-D-22-14026A meta-epidemiological assessment of transparency indicators of infectious disease modelsPLOS ONE

Dear Dr. Ioannidis,

Thank you for submitting your manuscript to PLOS ONE. After careful consideration, we feel that it has merit but does not fully meet PLOS ONE’s publication criteria as it currently stands. Therefore, we invite you to submit a revised version of the manuscript that addresses the points raised during the review process.

We look forward to receiving your revised manuscript.

Kind regards,

Chi-Hua Chen, Ph.D.

Academic Editor

PLOS ONE

Journal Requirements:

"I have read the journal's policy and the authors of this manuscript have no competing interests"

Reviewers' comments:

Reviewer's Responses to Questions

**Comments to the Author**

1. Is the manuscript technically sound, and do the data support the conclusions?

Reviewer #1: Yes

2. Has the statistical analysis been performed appropriately and rigorously? 

Reviewer #1: Yes

3. Have the authors made all data underlying the findings in their manuscript fully available?

Reviewer #1: Yes

4. Is the manuscript presented in an intelligible fashion and written in standard English?

Reviewer #1: Yes

5. Review Comments to the Author

Reviewer #1: This is a very interesting and thought provoking article. It is a common sentiment that scientific rigour and due process has been sacrificed during the course of the pandemic over a perceived notion of urgency, or sometimes over the need of beating others to the punch and claiming the glory. Not surprisingly, mathematical models of infectious diseases became darlings of the main steam media due to their striking, often sensational, conclusion which were very suitable as headlines. More importantly, they were often used to guide public health policies that had impact and will have lasting consequences on the society. I congratulate Dr. Ioannidis and Zavalis on their work and my opinion of the paper is positive.

The search strategy appears to be robust and they opted to screen the retrieved citations with a single author. This is rather unconventional but unlikely to change the outcomes but it would be worthwhile to explain why they opted to do so.

Using an R package to extract the information is wise and likely to reduce any inadvertent mistakes. Verification of the automated extraction with random checks is also good practice. I would have personally opted to verify any positive findings for variables in which the algorithms showed less than 95% PPV but complete random checks are also fine and a different methodology is unlikely to change the conclusions.

Findings are self explanatory and unfortunately, quite damning. It is inconceivable that a mathematical model would be published without open access codes and I would not mind if a stronger language was used to criticise these points. While Dr. Ioannidis stated pre-registration may be cumbersome, I believe it is entirely feasible and several items (assumptions, data sources to be used, general structure of the model...) can and should be made public before modeling begins so readers can appreciate the evolution of the algorithm whether it was natural or forced to achieve a specific end.

A mathematical model SHOULD NOT be taken seriously, talked about or published WITHOUT the underlying code and data sources are made public. This includes the models published by academic journals or unpublished ones that were used to guide health policies alike.

6. PLOS authors have the option to publish the peer review history of their article (what does this mean?). If published, this will include your full peer review and any attached files.

Reviewer #1: **Yes: **Erkan Kalafat

---

## [Author Response · Author response to Decision Letter 0]

19 Jul 2022

It is outlined in the Response to Reviewers.docx file.

---

## [Decision Letter · Decision Letter 1]

12 Aug 2022

PONE-D-22-14026R1A meta-epidemiological assessment of transparency indicators of infectious disease modelsPLOS ONE

Dear Dr. Ioannidis,

Thank you for submitting your manuscript to PLOS ONE. After careful consideration, we feel that it has merit but does not fully meet PLOS ONE’s publication criteria as it currently stands. Therefore, we invite you to submit a revised version of the manuscript that addresses the points raised during the review process.

We look forward to receiving your revised manuscript.

Kind regards,

Chi-Hua Chen, Ph.D.

Academic Editor

PLOS ONE

Reviewers' comments:

Reviewer's Responses to Questions

**Comments to the Author**

1. If the authors have adequately addressed your comments raised in a previous round of review and you feel that this manuscript is now acceptable for publication, you may indicate that here to bypass the “Comments to the Author” section, enter your conflict of interest statement in the “Confidential to Editor” section, and submit your "Accept" recommendation.

Reviewer #2: (No Response)

2. Is the manuscript technically sound, and do the data support the conclusions?

Reviewer #2: Yes

3. Has the statistical analysis been performed appropriately and rigorously? 

Reviewer #2: No

4. Have the authors made all data underlying the findings in their manuscript fully available?

Reviewer #2: Yes

5. Is the manuscript presented in an intelligible fashion and written in standard English?

Reviewer #2: Yes

6. Review Comments to the Author

Reviewer #2: This review measures the frequency of several indicators of scientific transparency in the infectious disease modeling literature. The authors study a sample of PMC papers from 2019 and 2021. They compare papers from 2019 to 2021, non-covid papers from 2019 to 2021, and non-covid papers from 2021 to covid papers from 2021.

This study addresses important and worthwhile research questions. Transparency is vital in mathematical modeling. Infectious disease modeling became far more prominent during the COVID-19 pandemic, and many stakeholders have emphasized the importance of transparency as this transition has taken place. Nevertheless, it is unclear how much these renewed calls for transparency have affected research practices, and examples abound of mathematical models failing to prove useful in policymaking and public health decision-making.

The authors build on previous work assessing transparency in biomedical research. They use an established R package to automate the extraction of transparency indicators. They include a concise and clear description of the manual coding process. They use an appropriate p-value for statistical tests. The discussion is appropriate given the results. Results are well framed in the literature on transparency in mathematical and infectious disease modeling. The limitations are mostly comprehensive and defensible.

Specific comments:

1. The manual validation process should include some negative examples as well as positive examples. The authors claim that this is unnecessary due to the transparency indicator extraction algorithm's high NPV in prior work. However, the PPV in this study was substantially lower than in their previous paper. The authors rightly acknowledge that this body of research is substantively different from the literature with which the algorithm was developed. Moreover, the conclusions of the paper largely rest on the shockingly low frequencies of many transparency indicators. Without manual review of negatives, however, it is uncertain how frequent false negatives may be.

2. The authors do not address the multiple comparisons problem in their statistical tests. While it seems unlikely that it would affect the results greatly, this study design is certainly subject to this problem. The authors should either adjust for multiple comparisons or defend their decision not to do so.

3. The regression analysis is not adequately described. In methods, the authors state "we would conduct multivariable logistic regressions as well", without describing any of the details of these models (IVs, DVs, etc). In results, the authors state "Multivariable regressions (not shown) showed similar results." Similar in what sense? Including these results could be more convincing than the Fisher's exact tests, since they may be less affected by the multiple comparisons problem. (This, of course, depends on the details of the analysis, which are not clear as written.)

4. There are some run-on sentences and grammatical errors in the manual validation section.

5. In limitations, the authors mention that the sample of PMC papers is not representative of all PubMed indexed papers. Much of the covid modeling literature has been published as preprints or in other non-indexed sources. It would be interesting to compare covid research published in such venues to peer-reviewed covid research. Worth mentioning in the limitations and perhaps a promising follow-up to this paper.

6. The first sentence of the conclusion is confusing. Should it read "…as data" rather than "… and data"?

The expectation of pre-registration when using models to make future predictions seems unrealistic. It is unclear to me how this would work. Even in such cases, modelers often don't know the final form the model will take when development begins.

7. PLOS authors have the option to publish the peer review history of their article (what does this mean?). If published, this will include your full peer review and any attached files.

Reviewer #2: **Yes: **Alexander Preiss

---

## [Author Response · Author response to Decision Letter 1]

30 Aug 2022

See response to reviewers document that has been attached.

---

## [Decision Letter · Decision Letter 2]

15 Sep 2022

A meta-epidemiological assessment of transparency indicators of infectious disease models

PONE-D-22-14026R2

Dear Dr. Ioannidis,

We’re pleased to inform you that your manuscript has been judged scientifically suitable for publication and will be formally accepted for publication once it meets all outstanding technical requirements.

Kind regards,

Chi-Hua Chen, Ph.D.

Academic Editor

PLOS ONE

Additional Editor Comments (optional):

Reviewers' comments:

Reviewer's Responses to Questions

**Comments to the Author**

1. If the authors have adequately addressed your comments raised in a previous round of review and you feel that this manuscript is now acceptable for publication, you may indicate that here to bypass the “Comments to the Author” section, enter your conflict of interest statement in the “Confidential to Editor” section, and submit your "Accept" recommendation.

Reviewer #2: All comments have been addressed

2. Is the manuscript technically sound, and do the data support the conclusions?

Reviewer #2: Yes

3. Has the statistical analysis been performed appropriately and rigorously? 

Reviewer #2: Yes

4. Have the authors made all data underlying the findings in their manuscript fully available?

Reviewer #2: Yes

5. Is the manuscript presented in an intelligible fashion and written in standard English?

Reviewer #2: Yes

6. Review Comments to the Author

Reviewer #2: I thank the authors for addressing my comments, particularly the manual validation of negatives. The conclusions stand despite a modest proportion of false negatives. The correction formula is a clever addition.

7. PLOS authors have the option to publish the peer review history of their article (what does this mean?). If published, this will include your full peer review and any attached files.

Reviewer #2: **Yes: **Alexander Preiss

---

## [Editor Report · Acceptance letter]

27 Sep 2022

PONE-D-22-14026R2 

A meta-epidemiological assessment of transparency indicators of infectious disease models 

Dear Dr. Ioannidis:

I'm pleased to inform you that your manuscript has been deemed suitable for publication in PLOS ONE. Congratulations! Your manuscript is now with our production department. 

Kind regards, 

on behalf of

Professor Chi-Hua Chen 

Academic Editor

PLOS ONE